# Stiff-Leg Syndrome Associated with Autoimmune Retinopathy and Its Treatment with IVIg—A Case Report and Review of the Literature

**DOI:** 10.3390/brainsci13101361

**Published:** 2023-09-23

**Authors:** Vassilis E. Papadopoulos, George K. Papadimas, Sofia Androudi, Maria Anagnostouli, Maria-Eleftheria Evangelopoulos

**Affiliations:** 1First Department of Neurology, School of Medicine, Eginition Hospital, National and Kapodistrian University of Athens, 11528 Athens, Greeceevangelopme@med.uoa.gr (M.-E.E.); 2Department of Ophthalmology, University of Thessaly, 41110 Larissa, Greece

**Keywords:** stiff-leg syndrome, GAD, autoimmune retinopathy, IVIg, stiff-person syndrome

## Abstract

Antibodies to glutamic acid decarboxylase (GAD) have been predominantly associated with stiff-person syndrome (SPS), which is often accompanied by organ-specific autoimmune diseases, such as late-onset type 1 diabetes. Autoimmune retinal pathology in SPS has recently been suggested to coexist in patients suffering from this disease; however, evidence reporting potential treatment options for the neurological and visual symptoms these patients experience remains scarce. We provide a review of the relevant literature, presenting a rare case of a middle-aged woman with autoimmune retinopathy (AIR) followed by stiff-leg syndrome who responded to intravenous immune globulin treatment (IVIg). Our report adds to previously reported data supporting the efficacy of IVIg in SPS spectrum disorders while also proposing the potential effect of IVIg in treating SPS spectrum patients with coexisting AIR.

## 1. Introduction

Antibodies to glutamic acid decarboxylase (GAD) have been associated with several neurological syndromes, predominantly including stiff-person syndrome (SPS), followed by cerebellar ataxia and epilepsy [1]. A “focal” form of SPS has also been described, with patients presenting with symptoms similar to SPS but confined to one leg. This syndrome is also associated with GAD antibodies and has been termed stiff-leg syndrome (SLS) [2,3].

Autoimmune retinopathy (AIR) is a rare syndrome, mostly associated with a coexisting cancer diagnosis, while also presenting as an isolated autoimmune disease with antibodies against optic nerve antigens [4]. An extremely uncommon presentation of AIR with serum antibodies against GAD and a coexisting SPS syndrome has been previously described [5,6]; however, it is speculated that retinal pathology may be a clinically silent or overlooked occurrence in SPS patients [7].

Immunological treatment, including intravenous immune globulin (IVIg), has been applied to both SLS and AIR patients, with mixed results [3,4,6].

We report a very rare case of a middle-aged woman presenting with AIR followed by SLS, who responded to IVIg treatment. Along with a review of the literature sporadically addressing these infrequent autoimmune neurological cases, our report aims to support the hypothesis of a more frequent than previously thought coexisting AIR in SPS patients. Notably, our work offers previously undescribed data suggesting the use of IVIg in the treatment of GAD-related AIR.

## 2. Stiff-Person Syndrome (SPS) Spectrum Disorders

SPS spectrum disorders, also described as GAD-antibody-related spectrum disorders, comprise a wide spectrum of diseases, including but not limited to SPS [1]. The characteristic clinical syndrome of rigidity, spasms and continuous central motor unit activity was first described in 1956 as stiff-man syndrome [8]. However, the spectrum of SPS disorders now ranges from minor involvement with symptoms limited to one leg [2,3] to potentially life-threatening progressive encephalomyelitis with rigidity and myoclonus [9]. These patients who had additional symptoms and signs on top of the classic phenotype of rigidity were initially described as SPS-plus patients [10]. The term has, however, been gradually abandoned in the literature as the spectrum of phenotypes has expanded and does not always involve an underlying SPS phenotype. Thus, the term GAD-antibody spectrum disorders is proposed [11].

### 2.1. Stiff-Person Syndrome (SPS)

SPS is more common in women, and the usual age of onset is 30 to 50 years old [12]. There is a predominance in the African American ethnicity [12]. Prevalence is estimated to be around 1 case per million people, coining SPS as a rare disease with various phenotypes of even lower prevalence [1].

The classic SPS clinical syndrome starts with proximal leg rigidity, followed by truncal muscle involvement [11]. The pattern of continuous motor activity undertaken by the agonist and antagonist muscles results in a board-like appearance of the abdominal wall, owing to the co-contraction of paraspinal and abdominal muscles, potentially aiding in diagnosis when investigated by electromyography (EMG) [1]. In extreme and severe cases, muscle spasms can be so intense that they result in bone fractures and joint subluxations [13].

Superimposed on muscle rigidity, patients also suffer from intermittent spasms in the truncal and proximal lower extremity muscle groups, probably in response to stimuli. A common finding from neurological examination also involves hyperreflexia [1,11]. Accompanying these motor symptoms, a common non-motor symptom of SPS patients is severe anxiety and task-specific phobias, which, more often than presumed, lead to misdiagnosis [1,11]. These symptoms cloud diagnosis when present, as patients might be unable to cooperate in a full neurological examination due to fear of walking in open spaces [1].

Patients with GAD-antibody-related syndromes typically have a concomitant organ-specific autoimmune disorder [1]. Around half of patients have a family history of autoimmunity, while about half of patients also have a DM diagnosis [14], which is the most frequently observed accompanying autoimmune disease. Other autoimmune disorders co-existing in GAD-antibody neurological patients include Hashimoto thyroiditis, Graves’ disease, vitiligo, and pernicious anemia [1,12].

The prognosis of SPS is, on the whole, unfavorable despite the application of various immunotherapies. In fact, around 40% of patients are unresponsive to immunotherapy, eventually reaching a non-ambulatory state within 5 years [15]. This, on the other hand, translates to the fact that most patients are able to remain ambulatory but usually require assistance due to disability [13]. The prognosis also differs between variants and depends on comorbidities, especially in the elderly [16]. What should be stressed in terms of prognosis is that the prompt initiation of therapy is of the utmost importance, as untreated patients tend to progress to a disabled state, refractory to treatment [17].

### 2.2. Progressive Encephalomyelitis with Rigidity and Myoclonus (PERM)

Although now recognized to be primarily correlated with antibodies targeting the GlyR receptor, PERM was initially considered an SPS-spectrum disorder [18]. PERM is a more severe, as well as distinct phenotype, which is characterized by stiffness, spasms, myoclonus, and brainstem dysfunction that present as eye movement disorders, dysphagia, ataxia, and prominent involvement of autonomic dysfunction that can progress to be fatal [9]. Adding to the hypothesis of a common disease pathway, the diagnostic hallmark of this entity, which is a GlyR-antibody reactivity, is also present in some SPS patients. In fact, about 15% of SPS patients are double positive for GAD and GlyR, albeit with low titers to the latter [19].

### 2.3. Cerebellar Ataxia

The GAD-antibody spectrum also includes, however, clinical syndromes beyond classic SPS. First described in 1997, patients with a GAD-antibody serum positivity can also develop cerebellar ataxia [20]. Although proven later to be a particularly rare entity, affecting about 2% of patients with sporadic cerebellar ataxia [21], anti-GAD antibody-associated cerebellar ataxia represents the second most frequent phenotype of GAD-related disease on the spectrum [11]. In line with the epidemiology of SPS, it chiefly affects women and is also accompanied by other organ-specific autoimmune disorders, such as DM [20]. This phenotype has a subacute onset and a progressive course and mostly involves gait ataxia [1]. Interestingly, around a quarter of patients with GAD-related cerebellar ataxia exhibit a predominantly vertigo phenotype, also involving eye symptoms in the form of diplopia [22], while some patients have a more severe eye movement disorder, ranging from nystagmus to opsoclonus, which can also be isolated [23]. Aiding in diagnosis, the magnetic resonance imaging (MRI) of such patients can infrequently reveal a mild cerebellar atrophy [24]; however, the underlying pathology is speculated to be largely a functional disorder [11].

### 2.4. Epilepsy and Limbic Encephalitis

Third in line among the most common phenotypes of GAD-antibody-related spectrum disorders, autoimmune epilepsy in the presence of GAD-antibody serum positivity mostly concerns refractory temporal lobe epilepsy [25].

In line with the syndromes presented above, epidemiological studies suggest that most patients with GAD-related autoimmune epilepsy are female. However, when compared to patients presenting with the SPS or cerebellar ataxia phenotypes, these patients are younger and seem to have a lower incidence of organ-specific autoimmune comorbidities [26,27].

As far as GAD-related autoimmune limbic encephalitis is concerned, a matter of diagnostic unclarity and a sense of overdiagnosis has lately been implied [1,11]. Criteria for the entity of limbic encephalitis are often not fulfilled, thus casting a shadow of doubt on whether this entity is indeed that common among patients with GAD-antibody-related epilepsy [1]. A small proportion of such patients fulfill, however, the above-mentioned criteria and seem to respond to immunotherapy [28]. Nonetheless, as is the case with all GAD-antibody-related autoimmune syndromes, the pathogenicity of GAD antibodies in limbic encephalitis patients is speculated to result from the concomitant presence of other pathogenic antibodies, most likely GABA_B_ receptor antibodies [11,29].

### 2.5. Paraneoplastic Syndromes

The correlation between a GAD-antibody-related neurological syndrome with an underlying tumor has been challenged by various researchers and clinicians. It is estimated that around 5% of SPS patients have a paraneoplastic GAD-antibody-related syndrome, with a concomitant positivity to other epitopes that are known to result in paraneoplastic syndromes, namely amphiphysin and gephyrin [30,31]. Patients presenting with a phenotype of SPS or cerebellar ataxia seem to have a higher possibility of having paraneoplastic syndrome. However, a syndrome that is not typically associated with GAD antibodies can also be present, e.g., limbic or brainstem encephalitis, opsoclonus-myoclonus with ataxia syndrome, or SPS confined to an upper extremity. Underlying tumors most frequently involve lung cancer and thymoma [1].

### 2.6. Stiff-Leg Syndrome (SLS)

A more “benign” phenotype of SLS, with rigidity limited to one leg, is an even rarer syndrome, scarcely described in the literature [2,3]. Patients develop a syndrome starting as a typical SPS syndrome, with an atypical course that never develops into a complete SPS syndrome. They seem to have a long duration of symptoms only limited to one or two legs, with stiffness and painful spasms persisting for years. Reports suggest that these symptoms remain confined to one or two extremities for up to 16 years [2,3]. EMG is similarly useful in diagnosis, revealing continuous motor activity and enhanced reflexes, as in SPS [2,3]. Patients with SLS also exhibit phobias and a high titer of serum GAD antibodies [3]. Symptomatic treatment options include baclofen with an eventual combination of diazepam, in a similar fashion to classical SPS [2], or botulinum toxin injections.

### 2.7. Stiff-Person Syndrome (SPS) Immunopathogenesis

The mechanism behind the pathogenesis of GAD-antibody-related syndromes has been thoroughly investigated; however, it remains, as yet, unclear [1,11,14]. GABAergic neurons comprise a large inhibitory network in the CNS as inhibitory interneurons [32]. They express GAD65 and are chiefly located in spinal gray matter, the hippocampus, the cerebellum, the basal ganglia, and brainstem nuclei [14]. While it has been speculated that GAD antibodies have a pathogenic role in related syndromes through their inhibitory effect on GAD65, eventually leading to reduced GABA synthesis and impaired GABAergic inhibitory transmission [14,33], this has not been sufficiently proved by related animal experiments and remains controversial [1].

Moreover, the fact that GAD is an intracellular antigen poses a question on the mechanism through which GAD antibodies affect GAD’s function in vivo. A hypothesis of the transient exposure of these intracellular epitopes during synaptic transmission has been hypothesized but not proven [11].

Nevertheless, a very high titer of GAD antibodies is of importance when diagnosing a GAD-related neurological syndrome, with a titer exceeding a value of 10,000 IU/mL considered through various clinical studies to be high enough [11,34]. These significantly high titers are considered sufficient to result in the intrathecal presence of GAD antibodies [23], deeming a lumbar puncture in the search for intrathecal GAD antibodies unnecessary [11]. Still, the importance of the intrathecal synthesis of GAD antibodies remains, and the so-called “CSF GAD antibody index” has been suggested as a diagnostic tool for patients suspected to have a neurological syndrome relating to GAD antibodies [35]. Using Link’s formula, one should use the ratio of the CSF GAD antibody titer to that in the serum, divided by the albumin concentration in the CSF and the serum, respectively. Values that are above one indicate intrathecal synthesis [1].

These high enough titers, together with a suspected epitope antigen specificity, are speculated to contribute to the fact that GAD antibodies are also present in DM patients but do not result in CNS syndrome [1]. Similarly, it has been contemplated that the variable recognition of epitopes is the underlying difference leading to various phenotypes among patients with GAD antibodies [1]; this, however, has not been supported by a number of studies [14].

### 2.8. Stiff-Person Syndrome (SPS) Treatment

The treatment of both classic SPS and SLS involves symptomatic therapy for rigidity and spasms and immunologic therapy [11]. The treatment rationale behind this approach is to reverse the impaired reciprocal GABAergic inhibition with agents enhancing GABA, while also seeking to limit the immunopathogenesis of SPS [16].

On the one hand, regarding GABA-enhancing agents, GABA_A_ agonists such as clonazepam or diazepam are chosen [1,11,16]. GABA_B_ agonists are also an option, with muscle relaxants such as baclofen or tizanidine shown to be effective [11,16]. Lastly, it has been proven that antiepileptics enhancing GABA synthesis can be facilitated [36] while botulinum toxin is reserved for patients with focal symptoms or those who are unresponsive to the above-mentioned antispasmodics [37]. However, immunotherapy is required as patients progress or do not respond to symptomatic treatment. This concerns both classical SPS patients, as well as SLS patients [1,2,3,11].

A proposed step-by-step approach beyond symptomatic treatment has recently been updated by Dalakas, starting with IVIg [16]. If successful, maintenance with IVIg infusions should be seized and repeated every four to six weeks. If no benefit is observed after three months, or IVIg is not tolerated, using rituximab has been suggested as a promising option [16]. Plasmapheresis is mostly reserved as an acute, adjunct treatment [38], whereas autologous hematopoietic stem-cell transplantation is considered a last-resort approach for severe cases [16]. Other immunotherapies that have been suggested through various anecdotal reports are of limited benefit. These include corticosteroids (which should be avoided as they can exacerbate concomitant DM) and other immunosuppressive agents such as azathioprine, methotrexate, cyclophosphamide, and mycophenolate mofetil, all without enough evidence to support their systematic use [16].

The most promising option of IVIg is believed to have a positive clinical response, resulting in a better prognosis. The use of IVIg in SPS spectrum patients is the only treatment option that has proved to be efficient through a randomized trial, where patients showed an improvement in their stiffness and daily activities [35].

## 3. GAD-Antibody-Related Eye Syndromes

GAD-antibody-related syndromes, especially when involving cerebellar ataxia, often include abnormal eye movements [34,39,40]. While a symptom that is often overlooked, as the accompanying symptoms of SPS and/or cerebellar ataxia are more severe and handicapping to the patient, it is mostly when these symptoms present isolated that clinical vigilance is provoked.

Among the different abnormal eye movements presented, the most common sign concerns downbeat nystagmus, which is considered to be due to the involvement of vestibular nuclei [34], followed by opsoclonus, as part of the opsoclonus, myoclonus, and ataxia syndrome [23].

Retinopathy in the presence of GAD-antibody reactivity, or an SPS spectrum disorder, has been previously reported in the literature, mostly in the form of sporadic case reports [4,5,6]. However, interest has recently been rising in identifying a possible coexisting AIR in SPS patients [7].

### 3.1. Autoimmune Retinopathy (AIR)

AIR has been mostly identified as a paraneoplastic syndrome accompanying cancer, melanoma, or lymphoma diagnosis [41]. It is, in fact, believed that AIR represents a class of autoimmune diseases that result from a cross-reactivity of autoantibodies. This disease entity thus presumably represents the reactivity of these autoantibodies against retinal or retinal-like antigens [41].

A typical AIR syndrome involves progressive, painless, or bilateral visual loss. Involvement can be asymmetric and includes photosensitivity, color vision changes, and night blindness [42]. Pathology involves a cone, rod, or diffuse photoreceptor dysfunction [4], potentially leading to a diagnostic dilemma between an AIR and a cone–rod dystrophy diagnosis. Identification of a serum antibody helps in constituting a diagnosis of AIR; however, this is rarely achieved in patients with a non-cancer-related syndrome [41]. Patients with AIR related to cancer, especially small-cell carcinoma, frequently have the presence of anti-recoverin in their serum [43].

### 3.2. GAD-Antibody-Related Autoimmune Retinopathy (AIR)

Visual loss in the context of the GAD-related neurological syndrome seems to be more prevalent than previously observed. A pattern of painless central visual loss, together with a mild inability to discriminate color and photosensitivity, is the clinical hallmark of such ocular involvement in patients suffering from a GAD-related syndrome [4,5,6]. These symptoms suggest pathology localized in the retina level as the main site of the disease process rather than optic nerve involvement in a syndrome best described as AIR [5].

Indeed, AIR in the presence of anti-GAD antibodies has thus been reported in a few case reports, paving the way to unveiling a potentially overlooked comorbidity in the ‘outskirts’ of the central nervous system [4,5,6].

The autoimmune nature of GAD-related AIR has been postulated to be due to the presence of both GAD as well as other antibodies targeting the retina. In line with the unclear pathogenicity of anti-GAD antibodies in SPS, it is unclear whether anti-GAD antibodies have a direct role in retinal disease in patients with GAD-related AIR [4]. It has been hypothesized that antibodies against recoverin, a protein involved in cancer-associated retinopathy (CAR) [44], or gephyrin [31], a protein localized in rat retina [45], could have a role in patients with mixed SPS and retinopathy phenotypes.

Nevertheless, the retina is, indeed, enriched with GABAergic neurons, whereas the amacrine cells of the inner plexiform layer could be susceptible to GAD antibodies themselves, as they have been shown to express GAD65 [5,46]. Indeed, the pattern of staining of the inner plexiform layer, accompanied by weaker staining of the outer plexiform layer, was described in an SPS patient with visual loss when the patient’s serum was used to stain macaque retina. Intriguingly, when authors used a commercially available monoclonal GAD65 antibody to stain the same macaque tissue, an almost identical pattern was revealed [5]. Thus, antiretinal antibodies in the serum of this SPS patient were reasoned to be identical to the commercial anti-GAD antibody.

Along the same lines, further data suggest that the anti-retinal role of GAD antibodies becomes available through a recent case report. A patient presenting with isolated painless and progressive visual loss with photosensitivity and positive GAD antibody serum reactivity also tested positive for a 40 kDa retina antigen, a 62 kDa retina antigen, and a 20 kDa optic nerve antigen. The 40 kDa retina antigen could possibly correspond to either rhodopsin or the 40 kDa CAR antigen [4].

Based on these reports, a recent cross-sectional study showed that SPS patients had lower ganglion cells and inner plexiform layer (GCIPL) thicknesses, inner nuclear layer thicknesses, and letter acuity scores compared to sex-matched healthy controls [7]. Intriguingly, the lower GCIPL and inner nuclear layer thicknesses were associated with SPS severity and serum anti-GAD antibody levels. This was a finding that was maintained even after excluding patients with concomitant DM to control for potential DM-related optic pathology. The role of optical coherence tomography (OCT) as a biomarker of SPS could potentially be at play in the future [7].

### 3.3. GAD-Antibody-Related Autoimmune Retinopathy (AIR) Treatment

Similar to the treatment of AIR not related to GAD antibody reactivity [41], as well as the treatment of SPS spectrum disorders [11], GAD-related AIR has mainly been managed with immunotherapy with modest results [4]. Several different immunotherapy approaches have been suggested in the literature concerning AIR. These include systemic and local corticosteroids, plasma exchange, IVIG, and autologous non-myeloblastive hematopoietic stem cell transplantation [41]. Out of these approaches, plasma exchange and IVIG have been used in a few case reports of GAD-related AIR [4,5].

Plasma exchange was used in a patient with isolated visual symptoms related to GAD-antibody positivity. The clinical endpoints used to evaluate treatment efficacy included visual acuity, visual fields, fundus autofluorescence, and electroretinogram findings. After six cycles of plasma exchange, the patient’s condition remained stable [4].

In a patient who suffered from visual loss in the context of GAD-related SPS, both intravenous corticosteroids and methotrexate failed to alleviate visual symptoms. This progressive eye disease resulted in blindness [5].

A patient presenting with PERM, visual loss, and serum GAD-antibody reactivity was treated with corticosteroids, cyclophosphamide, and azathioprine with no response in terms of her visual symptoms. She was subsequently managed with monthly intervals of IVIg at a dose of 0.4 gr/kg/day for 5 consecutive days. After the first cycle of treatment, the patient showed a dramatic improvement in their ataxia symptoms but failed to show an improvement in their vision [6].

## 4. Case Presentation

A 46-year-old woman with an insignificant previous medical history developed a gradual visual acuity loss in her left eye (OS) accompanied by photosensitivity. Her family history was unremarkable for any neurological or ophthalmologic disorder. Her daughter was diagnosed with Hashimoto’s thyroiditis. She underwent an extensive ophthalmological examination. Electroretinography and genetic testing supported a diagnosis of cone–rod dystrophy, which confirmed retinopathy but failed to unveil a specific underlying mechanism.

After a course of 6 months, she developed a gradual retinopathy in the right eye (OD). An extensive ophthalmological evaluation was again unable to identify an underlying cause. Two years later, at the age of 49, she experienced a gradual rigidity in her right leg that caused her gait to become unstable, resulting in occasional falls. More specifically, the patient reported that her last fall happened when she was accidentally pushed by a child on the street and lost her balance due to the stiffness of her right leg. She was examined by a neurologist. The neurological examination revealed a visual acuity of 20/400 in OD and OS, an increased tonus on the right leg, and increased tendon reflexes on the same side. Her cognitive function was not affected. Her coordination was intact, and her fine motor movement in all other extremities was normal. Her eye movements were unaffected. The rest of the neurological examination was unremarkable. Imaging studies of the brain and spine were normal. The dopamine transporter scan was normal. Nerve conduction studies were normal, while EMG revealed the spontaneous, continuous contraction of both agonist and antagonist muscles in the right leg. The diagnosis remained inconclusive.

After two more years, she was reevaluated by a neurologist due to the now disabling rigidity of her right leg. Thorough immunological testing was conducted. Systemic autoimmunity antibodies were negative. Antibodies against GAD were highly positive, with a titer of 1,031,423 IU/mL. Anti-GAD antibody titers were measured with an ELISA assay (Euroimmun) set up at the Neuroimmunology Unit, University of Athens, as described [47]. A diagnosis of SLS was made, and the patient was admitted to our department for further assessment. Ophthalmologist re-evaluation disclosed a visual acuity of 20/400 in both eyes. Optical Coherence Tomography (OCT) showed atrophy in the ellipsoid zone and the entire posterior pole (Figure 1). A visual field plot examination revealed severely affected visual fields in both eyes (Figure 2A).

As suggested previously, a high titer of serum anti-GAD positivity prompted the further evaluation of a potential GAD-related neurological syndrome [11]. More importantly, when a serum titer measured using ELISA exceeded a value of 10,000 IU/mL, this was indicative of a neurological autoimmune syndrome relating to GAD antibodies [11,34]. An intrathecal investigation is not exclusively needed in such a case; however, we deemed it relevant to repeat an examination of serum GAD antibodies along with cerebrospinal fluid anti-GAD evaluation. Our goal was to further support the diagnosis of a neurological syndrome owing to GAD antibodies by proving the intrathecal synthesis of antibodies using the proposed GAD ab index [35]. The serum anti-GAD antibody titer was 210,801 IU/mL, and the cerebrospinal fluid (CSF) anti-GAD antibody titer was 14,762 IU/mL with a CSF GAD Index [11] of 11.8 (>1). The CSF IgG Index was 0.5, while CSF oligoclonal bands were positive for the T2 type. On the front of AIR, we sought to investigate whether the patient tested positive for antibodies previously shown to be present in AIR patients. Recoverin antibodies, previously described in patients with paraneoplastic retinopathy or AIR [44], were negative.

The patient was initially treated with baclofen at a dose of up to 50 mg per day in three divided doses, gabapentin at 300 mg three times daily and diazepam at 5 mg once daily. She showed a mild improvement in the rigidity of her right leg and the accompanying pain. She was subsequently treated with monthly infusions of IVIg with a dose of 2 gr/kg divided into five daily doses of 0.4 gr/kg. After 6 months of treatment, the patient reported an improvement in their optical perception, and visual acuity was tested at 20/200 in both eyes. An improvement was evident in the visual fields plot (representing functional improvement), as shown in Figure 2B. The rigidity of her right leg also showed a mild improvement. The OCT test was repeated, showing no significant anatomic improvements.

## 5. Discussion

GAD antibody-spectrum disorders entail a wide spectrum of clinical entities with overlapping symptomatology [1,11]. Beyond the classical phenotype of SPS with muscle rigidity, intermittent spasms and hyperreflexia alongside GAD-antibody-related disorders are now known to involve less frequent, partial SPS syndromes like SLS, as well as distinct clinical entities. GAD-antibody related cerebellar ataxia is the second most common presentation of GAD-antibody-related diseases, with patients presenting with gait ataxia, dysarthria, and eye movement disorders mostly in the form of downbeat nystagmus. The third most common syndrome in the GAD-antibody-related spectrum is autoimmune epilepsy, with a typical syndrome concerning refractory temporal lobe epilepsy. With or without the cerebellar syndrome of gait ataxia, patients with a GAD-antibody-related syndrome can also have visual symptoms. These symptoms are typically due to an underlying nystagmus or a related eye movement disorder, such as opsoclonus. Yet, the spectrum of GAD-antibody-related syndromes has lately been linked to visual symptoms owing to an underlying retinopathy. Patients suffering from such a disease present with painless visual loss, either isolated or accompanied by one of the above-mentioned neurological syndromes. These entities are difficult to diagnose, as awareness of their existence is lacking. Moreover, the severity of the accompanying ambulatory and psychological symptoms results in visual symptoms being overlooked despite their severity, which is typically handicapping.

All these entities are often accompanied by coexisting organ-specific autoimmune diseases, mostly latent autoimmune DM, thyroiditis, and pernicious anemia [1]. The wide spectrum of these phenotypes has been speculated to result from two facts. First, due to GAD antibodies showing an affinity to variable epitopes [48], as well as the fact that other antibodies might also be present in the serum or CSF of patients with SPS [11]. Indeed, anti-GAD positive patients frequently test positive to additional antibodies, targeting, among others, the glycine-a1 receptorresulting to PERM, GABA_A_ receptor, resulting to an autoimmune epilepsy syndrome, and gephyrin [11,31], a protein localizing in the rat retina [45], which causes visual symptoms in the form of painless visual loss.

Treating patients with GAD-related neurological syndromes is a challenging task. The link between GAD autoimmunity and SPS has paved the way for using immunomodulatory treatments in the therapy of these patients with mixed results [11]. Among many different immunomodulatory agents used historically, IVIg has been shown to be most effective at relieving stiffness and disability [35]. In the scarce literature reporting AIR in the presence of GAD antibody positivity [4] or SPS-spectrum disease [5,6], IVIg, intravenous corticosteroids, methotrexate, cyclophosphamide, azathioprine, and plasma exchange have been used, all proving ineffective when treating the visual symptoms of patients. The visual pathology of these patients seems to be progressive and eventually leads to blindness; however, patients were, in most cases, diagnosed with AIR with a significant delay. This suggests that an earlier and more aggressive treatment approach could be effective at halting disease progression. This pattern is in line with the treatment course of neurological symptoms of stiffness and/or ataxia in patients with more common phenotypes that mostly respond when treated early.

In our case, based on previous reports showing the efficacy of IVIg in the treatment of SPS [35] and SLS [3], we chose to treat our patient with IVIg. After a 6-month course of monthly IVIg infusions, a modest but remarkable improvement in her symptoms, both in visual fields and rigidity, was observed.

Our case highlights the importance of applying immunomodulatory treatment in patients presenting with GAD-related syndromes. Furthermore, it offers further clinical evidence of the previously speculated coexistence of retinal pathology in patients with SPS [7], strengthening the hypothesis that AIR can be part of a GAD-related syndrome, a hypothesis that has previously mostly been supported by histological and electrophysiological data [5,7].

Albeit the varying responses described in the literature, both in the efficacy of immunomodulatory treatment and, more specifically, in using IVIg to treat patients’ neurological symptoms, our work supports IVIg treatment, as a treatment option for these patients. Our report adds to existing data supporting IVIg as the first-choice treatment for GAD-related disease while also offering modest evidence of IVIg as the first-choice treatment in patients presenting with GAD-related AIR.

The search for potential therapeutic choices for accompanying visual symptoms of SPS patients, which seem to be frequently overlooked, seems to be the way forward for SPS patients. Whether immunomodulatory treatment is able to halt the visual part of disease progression when started earlier and more aggressively remains to be seen. Data from previous case reports, as well as in our case, have, up until now, been rather discouraging. It feels not over-optimistic, however, to argue that our change in focus alone, with raised awareness of this comorbidity in SPS and GAD-related disease patients, could lead us to paths yet unvisited. Using retinal pathology as a biomarker of disease burden and eventually measuring retinal layer thicknesses through OCT could prove useful in the future. An unmet need to provide more precise clinical guidance and more efficient therapeutic strategies for these patients could be a few steps away from being fulfilled.

## Figures and Tables

**Figure 1 brainsci-13-01361-f001:**
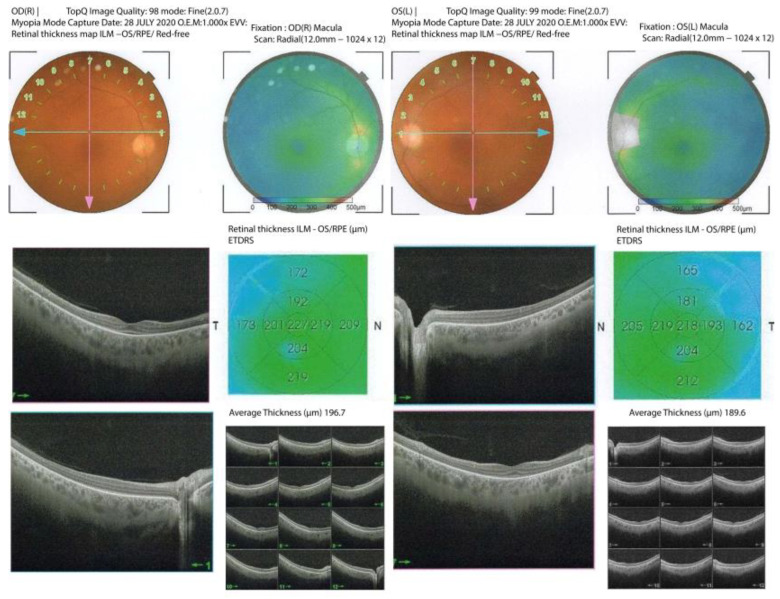
**Optical Coherence Tomography (OCT)** Optical Coherence Tomography of both eyes before IVIg treatment. The OCT exam remained the same after IVIg treatment.

**Figure 2 brainsci-13-01361-f002:**
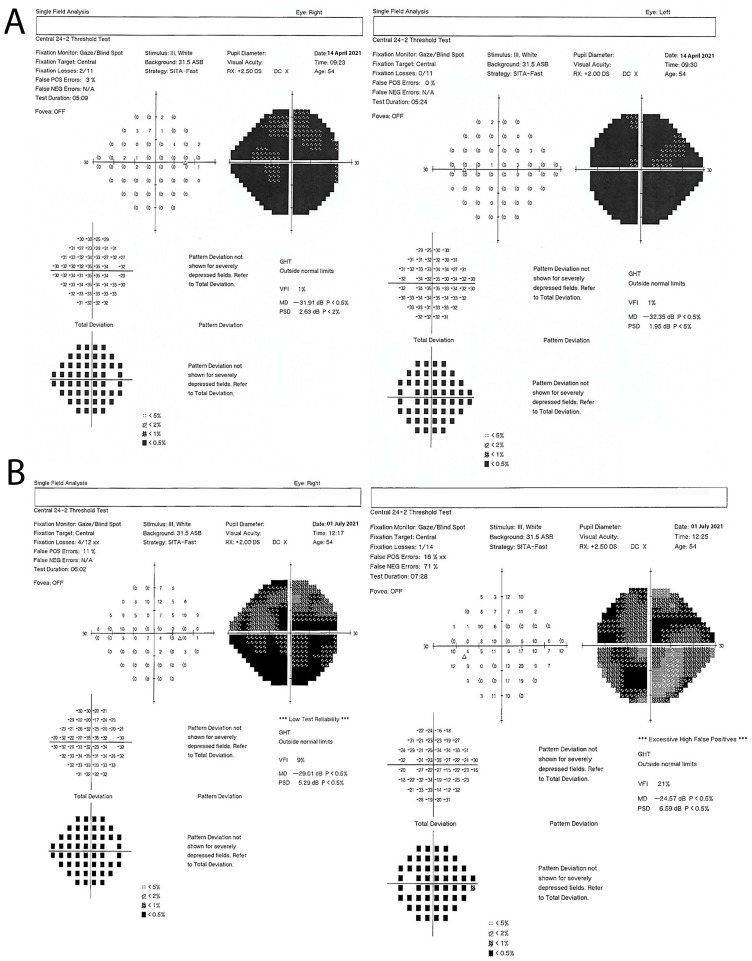
**Visual field plot.** Visual field plot exam of both eyes before ((**A**), **top**) and after ((**B**), **bottom**) IVIg treatment. A significant improvement was obvious.

## Data Availability

Data sharing is not applicable.

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
