# Peer review of "Stiff-Leg Syndrome Associated with Autoimmune Retinopathy and Its Treatment with IVIg—A Case Report and Review of the Literature"

_brainsci, 2023, doi:10.3390/brainsci13101361_

Round 1

Reviewer 1 Report

Dear Authors,

Thank you for interesting case report and review of SPS spectrum disorders.

I suggest some changes within your paper:

1. Describe the concept of "SPS plus" syndrome

2. Please, add a paragraph on pathological mechanisms of SPS/SLS to make the review more complete

3. There are more therapeutical options for SPS/SLS than just benzodiazepines and IVIG. Please, add some information on that

4. I would suggest adding information about prognosis in SPS

5. For the case report: Was visual evoked potentials performed? If so, what was the result? 

Author Response

We would like to thank the reviewer for taking the time to review this manuscript. Here follow our replies to their comments and suggestions.

  1. We have added a short description of the use of “SPS plus” term, now broadened and more frequently described as “GAD-antibody related” syndrome (lines 49-53).
  2. We thank you for this remark and agree that such a section was needed, as also observed by reviewer #2. We have thus added section 2.7, found in lines 152-182.
  3. We have now expanded section 2.8, going more vigorously through the different treatment options in SPS/SLS (lines 183-210)
  4. We have added a paragraph commenting on prognosis (lines 78-85).
  5. We thank you for this comment. Visual evoked potentials were not performed on the patient, as the patient was early on diagnosed with retinopathy, deeming such an examination irrelevant. The retinopathy was severe by the time she developed SLS, which was when she finally reached to neurology and our department. We have, as per other reviewers’ comments, added figures of OCT and visual fields plan examinations of the patients, aiding in the more complete presentation of this case report. We do not include the post-IVIg Treatment OCT for space purposes since it was practically unchanged.

Reviewer 2 Report

The thought behind this paper is good and the authors stated an important clinical correlate of autoimmune retinopathy in the setting of high GAD antibody titres. However, I have a few concerns about the manuscript that the authors should address. There is a lot of discussion about the epidemiology of the various types of syndromes associated with GAD-antibodies; however, the same level of discussion of the mechanisms involved in the immune and pathological processes is lacking. The authors stated that OCT has a potential of being a biomarker for SPS. Why is there no OCT result(s) included in this case report? Do the authors have any electroretinogram for this patient? Do the authors have any visual field plot for this patient? If yes, provide a before and after visual field plots, electroretinogram, and OCT.

Line 51: The authors should check that this cited reference relates to and support the claim made, as the cited reference appear to be misplaced. 

Lines 79 – 81: The authors should reword this sentence to render it more comprehensible.

Lines 222 – 225: The authors should reword this sentence to render it more comprehensible.

Author Response

We thank the reviewer for their time and commentary, as well as their suggestions.

  1. We agree that a section on immunopathogenesis was lacking and have added section 2.7 (lines 152-182).
  2. We have now accordingly supplemented the case presentation with OCT results, together with visual fields plots before and after treatment (Figures 1 and 2; we do not include the post-IVIg Treatment OCT for space purposes, since it was practically unchanged). We are of the impression that electroretinogram results exceed the scope of this work and thus decided not to include them in the final manuscript. However, electroretinogram examination was performed, with her most recent multifocal ERG showing decreased amplitude in both eyes (Right Eye: 32Nv/deg and Left Eye: 44Nv/deg). A multifocal ERG post-IVIg treatment is not available.
  3. Thank you for this remark. The text previously in line 51 is now citing the correct reference (line 55, reference 12).
  4. The two exempts cited by the reviewer have been edited and are now more eloquently phrased (lines 93-95 / 282-286).

Reviewer 3 Report

Papadopoulos and colleagues describe a case of Stiff Leg Syndrome associated with autoimmune retinopathy (AIR), with a review of related literature. The authors hypothesize the causative role of anti-GAD antibody in patient’s retinopathy and offer therapeutic strategy in this context.

 Pathogenic antibodies against retinal proteins known to be associated with visual loss due to autoimmune retinopathy were absent or not tested for this report. No additional experiments with the patient’s serum were performed in order to elucidate possible targeting of retinal antigens by GAD antibody.

 This paper expands observations on important visual disturbances encountered in Stiff Person Syndrome Spectrum disorders, but it does not provide evidence or plausible explanation for a possible GAD antibody pathogenic role in autoimmune retinopathy.

Author Response

We would like to thank the reviewer for taking the time to review this manuscript.

On the front of antibodies associated with autoimmune retinopathy (AIR), either neoplastic or paraneoplastic, we were able to test for recoverin, one of the most frequently associated antigens in AIR (Comlekoglu et al, 2013). This was possible using commercial immunoblot testing for various paraneoplastic syndromes, including the recoverin antigen. We were unfortunately not able to further test for other retinal antigens that have been associated with AIR. We agree with the reviewer that it would be valuable to further test the patient’s serum, potentially also using retinal tissue. Unfortunately, this exceeds our lab’s and our collaborating labs’ current capacity.

We remain of the impression that a report of a patient with the coexistence of AIR and SLS, responding to immunotherapy -now also documented with illustrations of ophthalmological examinations found in Figures 1 and 2- is of great value.  Together with previous reports, discussed thoroughly in our review, our work lays the groundwork for future larger case series. Whether GAD antibodies will be finally unveiled to be pathogenic, either in SPS or GAD-related AIR, is a question that has been famously hard to resolve and remains to be answered.

Reviewer 4 Report

This is a report of one patient who the authors have see with a minor stiff person syndrome spectrum, mainly stiff limb, and retinopathy.  The case report itself is rather minimal and there are no figures to illustrate, for instance, the OCT changes and visual fields etc before and after the IvIg treatment that seems to have been effective.

The rest of the report is a review of SPS-SDs and although adequate as a short review, is hardly appropriate in the context of this case.  There are one or two things that are questionable - eg.  on line 68 they say that the vasst majority of SPS patients have other autoimmune disorders.  In this reviewer's experience it might be about 50% - what do they mean by vast?

One thing that is important and missing is to discuss whether the GAD antibody titres are important and whether they consider a specific cut-off to distinguish (possibly) those patients with diabetes from those with definite CNS condition.   Their patient did have very high titres initially, one million, but they were much lower, 200 thousand, in the paired serum/CSF analysis.   Was this  a real change or just technical error?   By which method were the serum and CSF samples tested?

Author Response

We thank the reviewer for their time and their suggestions to improve this manuscript.

  1. We acknowledge the fact that our work was lacking figures to illustrate the rather encouraging results of immunotherapy in this patient. We have now included Figures 1 and 2, with results of the OCT and visual fields plot respectively.
  2. We believe that an updated review of the spectrum of SPS is informative in this case, which, in itself, expands the spectrum. On the individual comments, we have now modified the excerpt previous on line 68, being more specific on the incidence of autoimmune comorbidities (lines 72-77).
  3. We thank the reviewer for this remark. GAD antibody titer and/or intrathecal synthesis of GAD abs is an important issue in the diagnosis of neurological patients with a possible GAD-related syndrome. We have updated this excerpt on the introduction (lines 166-182), as well as in the case presentation (lines 325-327, 339-349). We have added the appropriate discussion on the matter, namely the proposed cut-off of 10,000IU/mL, as well as the role of GAD antibody index. As for the reviewer’s question on the matter of a fluctuating titer, we have added the method used (ELISA), which was the same at both time points. The fluctuating titer observed between the timepoints was without a clinical significance as it remained well above the cut-off value of 10,000IU/mL. Besides, it has been previously observed that a titer can be fluctuating during the disease course, not being correlated with clinical fluctuations (Dalakas, Rakocevic, Dambrosia, Alexopoulos, & McElroy, 2017; Rakocevic, Raju, & Dalakas, 2004).

Round 2

Reviewer 2 Report

I am satisfied with your response to my questions/comments.

Reviewer 3 Report

The authors have addressed reviewers' questions and concerns appropriately.  The manuscript has been sufficiently improved and provides useful clinical information. 

Reviewer 4 Report

Thank you for trying to address the concerns I raised.